# Monsoon weather and early childhood health in India

Anna Dimitrova[1]*, Jayanta Kumar Bora[1,2]

**1** Wittgenstein Centre for Demography and Global Human Capital (IIASA, VID/ ÖAW and WU), International Institute for Applied Systems Analysis, Laxenburg, Austria, **2** Indian Institute of Dalit Studies, New Delhi, India

* anna.dimitrova@iiasa.ac.at

**Data Availability Statement:** The data underlying the results presented in the study are available from the Demographic and Health Surveys (https://dhsprogram.com/data/available-datasets.cfm) and the Climatic Research Unit at the University of East Anglia (http://www.cru.uea.ac.uk/data/).

## Abstract

### Background

India is expected to experience an increase in the frequency and intensity of extreme weather events in the coming decades, which poses serious risks to human health and well-being in the country.

### Objective

This paper aims to shed light on the possible detrimental effects of monsoon weather shocks on childhood undernutrition in India using the Demographic and Health Survey 2015–16, in combination with geo-referenced climate data.

### Methods

Undernutrition is captured through measures of height-for-age, weight-for-height, stunting and wasting among children aged 0–59 months. The standardised precipitation and evapo-transpiration index (SPEI) is used to measure climatic conditions during critical periods of child development.

### Results

The results of a multivariate logistic regression model show that climate anomalies experienced in utero and during infancy are associated with an increased risk of child undernutrition; exposure to excessive monsoon precipitation during these early periods of life elevates the risk of stunting, particularly for children in the tropical wet and humid sub-tropical regions. In contrast, the risk of stunting is reduced for children residing in the mountainous areas who have experienced excessive monsoon precipitation during infancy. The evidence on the short-term effects of climate shocks on wasting is inconclusive. We additionally show that excessive precipitation, particularly during the monsoon season, is associated with an increased risk of contracting diarrhoea among children under five. Diseases transmitted through water, such as diarrhoea, could be one important channel through which excessive rainfall increases the risk of stunting.

**Funding:** The International Institute for Applied Systems Analysis (IIASA) funded the publication of this article. IIASA encourages and actively supports its researchers to publish their research in journal articles or books that are made available for free to all users (gold open access). This project has received funding from the European Union's Horizon 2020 research and innovation program under grant agreement No 741105. Project Name: The Demography of Sustainable Human Wellbeing, EmpoweredLifeYears. The funders had no role in study design, data collection and analysis, decision to publish, or preparation of the manuscript.

**Competing interests:** The authors have declared that no competing interests exist.

## Conclusions

We find a positive association between childhood undernutrition and exposure to excessive monsoon precipitation in India. Pronounced differences across climate zones are found. The findings of the present analysis warn of the urgent need to provide health assistance to children in flood-prone areas.

## Introduction

The increase of global surface temperature has changed rainfall patterns across the globe, with some regions becoming increasingly arid while others receiving abnormal levels of precipitation [1], [2]. In South Asia, monsoon rainfall has become more erratic and, as a result, an increase in the incidence of extreme weather events, both droughts and floods, has been observed [3], [4]. This has direct implications for the health and well-being of populations in the region [5]. Prolonged droughts and flash floods can likewise compromise food security, decrease water quality, threaten economic livelihoods, and increase the transmission of communicable diseases, among other risks [6].

Exposure to climate shocks can be particularly harmful to young children. Reduced food intake and contraction of infectious diseases make children under five years of age most susceptible to undernutrition. It can result in having low weight for one's age (being underweight), low height for one's age (being stunted), low weight for one's height (being wasted) or deficiency in vital nutrients [7]. Not only are households likely to reduce food consumption during an economic hardship, but also to redistribute resources between household members and forego medical expenses [8], [9], all of which can disproportionately affect children. The consequences of childhood undernutrition can stretch well beyond the first years of life and translate into poor health and socio-economic outcomes during adulthood [10–12]. It is therefore important to understand whether climate change poses threat to children's health, and if so, where assistance is most needed.

India is a high-risk country due to its large population size, economic underdevelopment and high susceptibility to weather extremes. In 2018 alone, some 8 million people were affected by droughts and 23 million by floods, and cumulative damages reached 4 billion US dollars [13]. The frequency and magnitude of extreme weather events in the country is projected to increase over the next decades due to changing monsoon patterns [4], [14], [15]. On the one hand, a gradual decline in monsoon circulation and rainfall has been observed in central India leading to dry spells. On the other hand, extreme rainfall events are on the rise and posing an increasing risk of flash floods. Central India is particularly affected, with a three-fold increase in widespread extreme precipitation events observed between 1950–2015 [14].

Even in the absence of extreme climate events, India has one of the highest rates of childhood undernutrition in the world—one in three children under the age of five has stunted growth and one in five is suffering from wasting [16]. While some progress has been achieved towards reducing childhood undernutrition in India, it has been slow compared to other fast-growing economies in the region. Climate change could further slowdown or even reverse this trend. However, it is not yet clear to what extent climate change poses risk to children and which extreme climate events pose higher risk, given that India has seen an increase in both dry and wet spells. It is likely that the effects will be unequally distributed across the different climate zones in the country, which range from arid in the west to tropical wet in the south.

To date, the evidence on the impact of extreme weather events on children's health is inconsistent and mostly derived from community-based surveys [17]. Most studies that have established a link between drought exposure and childhood undernutrition focus on sub-Saharan Africa [18–22]. Crop failure and income shocks are the most common mechanisms used to explain this association. In the context of India, there is limited research linking droughts and children's health. A recent study based on nationally representative survey data for India reported that children under five who experienced a drought in utero or at birth had a higher probability of being underweight and severely underweight [23]. Another study, focusing on a drought-affected desert district in western Rajasthan, India, found evidence of growth retardation and protein-energy malnutrition in children under the age of five who had been exposed to a drought [24].

There is a larger number of studies that have established a link between flood exposure and negative child health outcomes in India and other South Asian countries. Rodriguez-Llanes et al. [25], [26], for example, report an increased risk of childhood undernutrition and diarrhoeal diseases among flooded communities in rural eastern India. Del Ninno and Lundberg [27] used panel data for Bangladesh and found that children aged under five who were exposed to the severe flood of 1998 had reduced growth potential even 15 months after the disaster. Gaire et al. [28] used cross-sectional data for Nepal and reported that floods increased the risk of moderate and severe stunting among children aged under five. Another paper reported that children born during the monsoon months in India had lower anthropometric scores than children born during the fall and winter months, relating this trend to the higher prevalence of diseases and food insecurity during the monsoon season [29]. Another study used the Household Hunger Scale (HHS), which is a household food security metric, in addition to child anthropometric status to detect climate-induced hunger in Bangladesh and Ghana [30]. The study identified a link between increased rainfall and hunger in Bangladesh. Nevertheless, others did not find a strong association between flood exposure and childhood undernutrition in Bangladesh [31] and diarrhoeal infections in India [32].

Based on the above, it cannot be concluded whether climate change is affecting children's health in India and which climate events pose higher risk. Given that monsoon rainfall is becoming more erratic and extreme weather events more common, further evidence is needed to uncover the potential hazards for children. Considering the large sub-national differences in climatic conditions and levels of economic development in India, it can be expected that the risks associated with climate shocks are not equally distributed across the country. More evidence is needed at the subnational level to understand the potential hazards for children posed by climate change.

This paper adds to the literature by analysing data from a nationally representative household survey for India and geo-referenced climate data. The survey data come from the Demographic and Health Survey (DHS), collected in 2015–16. The focus of our analysis is on children under five years of age at the time of the interviews. We use the standardized precipitation and evapotranspiration index (SPEI), which combines data on precipitation and potential evapotranspiration due to temperature, to capture weather variability. The two datasets are matched at the grid-cell level using the GPS coordinates of household clusters in the DHS data. Potential short-term effects of SPEI shocks on childhood health are captured through measures of weight-for-height, wasting, and severe wasting, while potential medium- to long-term effects are captured through measures of height-for-age, stunting, and severe stunting. The use of geo-referenced climate data and nationally representative survey data is a major advantage of this paper. Moreover, SPEI allows us to study the effects of both dry and wet climate anomalies on child undernutrition, an approach which is rarely employed in the literature.

We additionally explore differences in climate-related vulnerabilities across climate zones in India, which span from tropical wet to mountain and arid. Finally, we investigate the role of diseases in mediating the effect of climate shocks on child undernutrition. Considering that a large share of urban households and the majority of rural household in the country still lack access to clean water and sanitation facilities [33], climate shocks can increase the incidence of infectious diseases. While floods may contaminate the living environment, droughts may force households to access water from unsafe sources.

The results of the present study can serve as a critical input to policy makers, researchers, and health professionals working to improve children's wellbeing in areas prone to climate shocks. In view of the global sustainable development agenda, we hope to draw attention on the potential obstacles to achieving the Sustainable Development Goal (SDG) of eliminating childhood malnutrition, which is a key to the success of other SDGs, in the areas of education and health for example.

The rest of the paper is organised as follows. Section 2 presents the data and methods employed in the study. Section 3 introduces the empirical strategy. The results are presented in Section 4 and sections 5 concludes.

## Data and methods

### Health data

We use the most recent DHS survey for India, better known as the National Family and Health Survey (NFHS), conducted in 2015–2016. The survey is coordinated by the International Institute of Population Sciences (IIPS) in Mumbai under the stewardship of the Ministry of Health and Family Welfare, Government of India. As with other DHS surveys, the focus of the NFHS is on fertility behavior, health and general welfare of women in reproductive age and their children. The sample is representative at the national and subnational level. A detailed description of the procedure and the data can be found in the national report [34]. The 2015–16 NFHS round contains a sample of 256,244 children under the age of five, from an overall sample of 601,509 households.

Following standard practice, we use anthropometric measures for height and weight of children aged 0–59 months collected during the interviews to construct indicators for undernutrition. More specifically, we compute z-scores for height-for-age (HAZ) and weight-for-age (WAZ) relative to the World Health Oragnisation's (WHO) growth standard medians for children of the same age [35]. Binary outcome variables are then constructed based on the z-scores. Children who are more than two standard deviations (SDs) below the HAZ and WHZ median for their age group are classified as stunted and wasted, respectively. Those who are more than three SDs below the HAZ and WHZ median are classified as severely stunted and severely wasted, respectively. Fig 1 shows the distribution of HAZ and WHZ z-scores for our sample population.

The above measures are widely used to assess children's health and nutrition status. Each capture different aspects of childhood undernutrition. Stunting, for instance, reflects the cumulative effect of undernutrition and infections since the child's birth and even in utero. It can thus indicate poor environmental conditions or other long-term restrictions to a child's physical development [7]. Severe stunting can impair not only the physical but also the mental development of a child, with long-lasting implications. Children who are severely stunted have been found to perform worse in school and have reduced intellectual capacity [10–12]. In adulthood, stunted women are likely to experience complications during labour and give birth to stunted children, which creates a 'vicious cycle' of malnutrition [7], [36]. Wasting, on the other hand, indicates acute weight loss during a short period of time, which could be the result

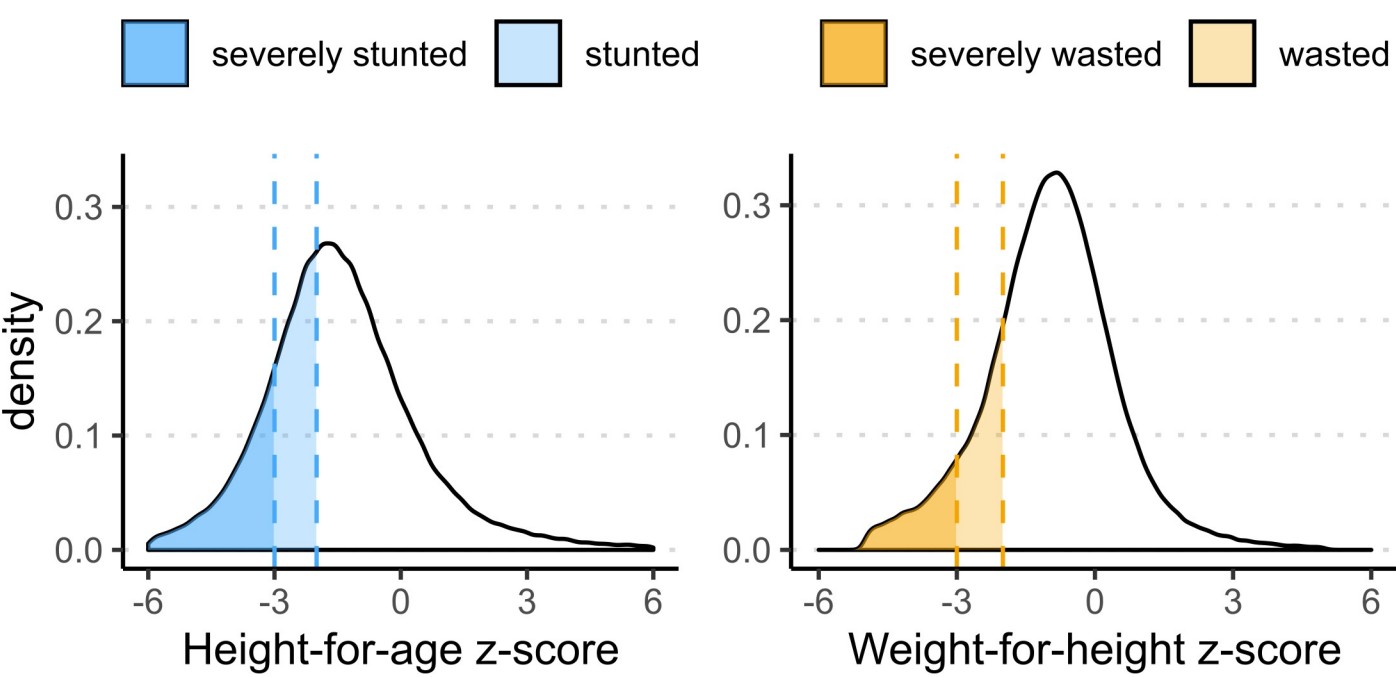

**Fig 1. Distribution of height-for-age (HAZ) and weight-for-height (WHZ) z-scores of children aged 0–5.** The vertical dashed reference lines indicate the -2 and -3 z-score thresholds which correspond with stunting (wasting) and severe stunting (severe wasting). Biologically implausible scores (HAZ < -6 or HAZ > 6; WHZ < -5 or WHZ > 5) were removed from the sample.

of low food intake or presence of diarrhoeal infections [7]. Undernourished children are also at a higher risk of death; 45% of all child deaths in low- and middle-income countries have been attributed to undernutrition [37].

The usefulness of reference values, such as HAZ < -2, for detecting linear growth deficits in children has been a subject of debate. At the individual level, applying an arbitrary cut-off point can lead to underestimation of health problems; Whether a child is slightly below or slightly above the cut-off line should not make big a difference to their health condition [38]. On a societal level, however, such cut-off points are informative as they allow the monitoring of child malnutrition over time [39], and reveal important socioeconomic inequalities when calculated for different population sub-groups [40], [41]. For the purpose of this analysis, HAZ and WHZ scores are used both as continuous variables and dichotomised using different cut-off points as described above.

One of the main mechanisms through which we expect climate shocks to affect child nutrition are diseases transmitted through water, such as diarrhoeal infections. The DHS surveys include self-reported information on diarrhoeal infections in children under five years of age. However, such incidences of diarrhoea are only reported in the two weeks preceding the survey. Therefore, it is not possible to establish a direct link between climate shocks, diseases and child undernutrition using DHS data. Nevertheless, we explore this pathway by assessing the relationship between climate variability around the time of the interview and the risk of diarrhoea among children under five.

### Climate data

We use the standardized precipitation and evapotranspiration index (SPEI) to measure monsoon season deviations in climatic conditions. SPEI measures monthly variations in the net values of precipitation minus potential evapotranspiration due to temperature compared to

the location-specific long-term mean. The index contains both negative values, indicating drought conditions, and positive values, indicating wet conditions. SPEI values close to 0 indicate near-normal conditions (see Table 1). Additionally, SPEI can be calculated at different time scales (from 1 to 48 months or more) to account for the cumulative effect of precipitation and evapotranspiration over previous months.

The SPEI index was developed in 2010 [42] and was originally intended for drought monitoring. It is an improvement on earlier drought indices as it allows using temperature along with rainfall data to measure the accumulation of water deficit/surplus. It is important to account for the effect of temperature on evapotranspiration in addition to rainfall. For example, in sunny places with high temperatures, excessive rainfall will quickly dry up, whereas in cooler and cloudier places, excessive water will accumulate and remain for longer, potentially affecting sanitary conditions. SPEI accounts for this in a way that rainfall alone does not. The SPEI index is increasingly used to monitor floods as well, especially when calculated at shorter timescales, such as 1 to 3 months [43], [44]. Floods are particularly difficult to forecast and show poor correlation with simple rainfall-based measures [45].

The SPEI index has the potential to improve both drought and flood forecasting. It can serve policy makers, particularly in countries with high risk levels but low levels of preparedness to such disasters. Global gridded SPEI data can be accessed online (https://spei.csic.es/database.html) or calculated with available software. Additionally, the index allows direct comparisons across time and space without the need to do additional standardisation.

We use the R package 'SPEI' to generate monthly SPEI values based on input precipitation and potential evapotranspiration data from the Climatic Research Unit's (CRU) time-series 3.25 [46]. The CRU data are available for the whole globe at 0.5˚ spatial resolution and cover the period from 1901 until 2016. We calculate 1-month SPEI values in order to capture both dry and wet anomalies.

We merge the climate and DHS data by using the geographic coordinates of household clusters, which are available in more recent DHS rounds. For confidentiality reasons, the location of households in the DHS data is shifted by 2 km for urban clusters and 5-km for rural clusters, with an additional shift of 10-km for 5% of all clusters [47]. To account for this shift, we create a 10 km radius around each cluster and average climatic conditions within this buffer area. This also accounts for the possibility that household may be affected by weather shocks in surrounding areas, for example if they travel some distance to farm or to collect water.

Additionally, the locations of household clusters are overlaid with a climate classification map to determine the predominant climate zone in the specific location. An updated Köppen-Geiger (KG) climate classification map is used for this purpose, available at 1-km resolution [48]. The KG system classifies climate into five main types and 30 sub-types based on air temperature and precipitation data and corrected for topographical effects. The KG maps are

**Table 1. SPEI range and classification.**

| SPEI range | Condition |
|---|---|
| SPEI $\leq$ -2 | Extreme drought |
| -2 < SPEI $\leq$ -1.5 | Severe drought |
| -1.5 < SPEI $\leq$ -1 | Moderate drought |
| -1 < SPEI $\leq$ 1 | Near normal |
| 1 < SPEI $\leq$ 1.5 | Moderately wet |
| 1.5 < SPEI $\leq$ 2 | Severely wet |
| SPEI > 2 | Extremely wet |

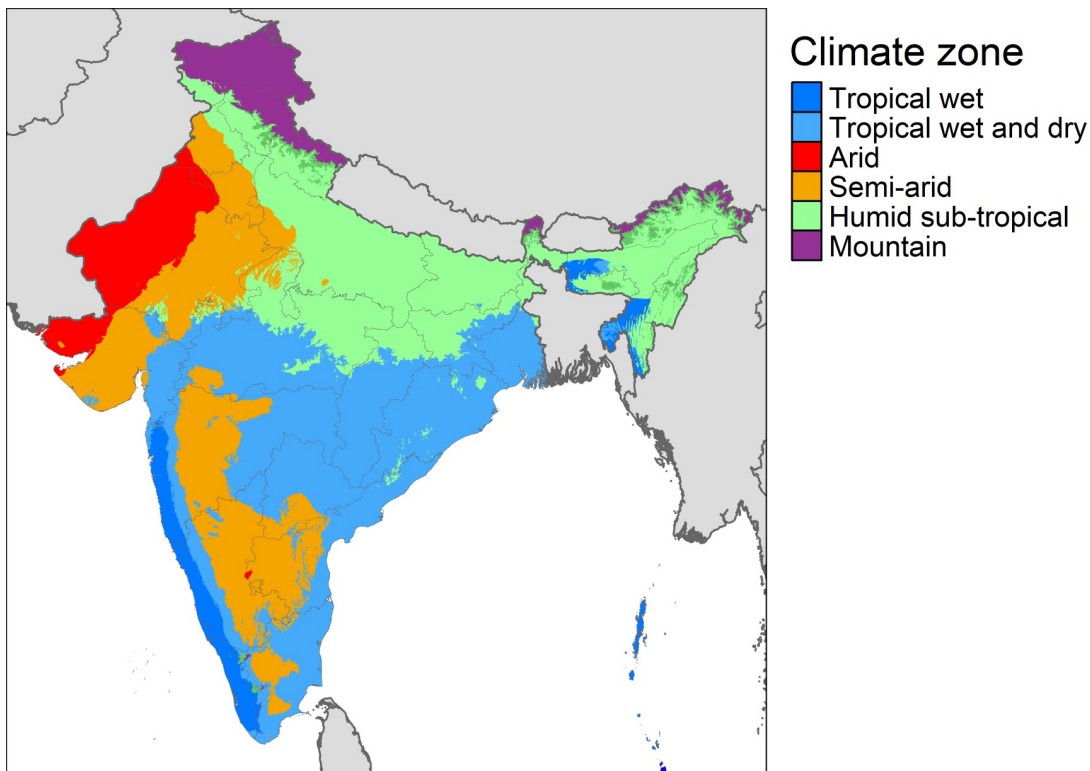

**Fig 2. Major climate zones in India based on Köppen-Geiger climate classification.** "Mountain" zone includes multiple climate regions in northern India. Map image is based on Beck, H. E. *et al.*, Present and future Köppen-Geiger climate classification maps at 1-km resolution. *Sci. Data*. 5:180214 doi: 10.1038/sdata.2018.214 (2018). The data used for producing this image is freely available under the Creative Commons license (accessed on 21.Feb.2020).

typically used in ecological models and climate change impact assessments [48]. As can be seen in Fig 2, India spans six main climate zones, ranging from tropical wet to arid.

The maps in Fig 3 show the substantial climatic differences across India. Mean annual temperatures around -5 degrees Celsius can be observed in the northern Himalayan region while temperatures of up to 30 degrees can be seen in the southern parts of the country. Annual rainfall is particularly high along the west coast, in the tropical regions of Kerala and Go, as well as in the north-eastern parts of the country. Large variations in SPEI can also be seen for the period 2009–2016, with northern and north-east India as well as the southern tip of the country becoming more arid, while central-west India becoming more humid.

For the purpose of this paper, we focus on climate conditions during the summer season (months June to September) when monsoon rains sweep across India. In each year, we average the monthly SPEI values from June to September in order to generate a measure of monsoon rainfall variability. About 75% of the annual rainfall in the country is received during the monsoon season [49] (see Fig 4B) and feeds into important economic activities throughout the rest of the year, such as power generation, agriculture and freshwater reservoirs. Nonetheless, monsoon rainfall can be very sporadic. Heavy downpours often lead to flash floods, while deficient and/or delayed monsoon rainfall can result in severe droughts.

Year to year variations in the timing and intensity of monsoon rainfall are becoming more common due to climate change [4], [14]. Central India has seen a particular increase in the frequency and intensity of extreme weather events, both droughts and floods [14]. The spatial variations in monsoon season SPEI can be seen in Fig 3 and Fig 4. Between 2009 and 2016,

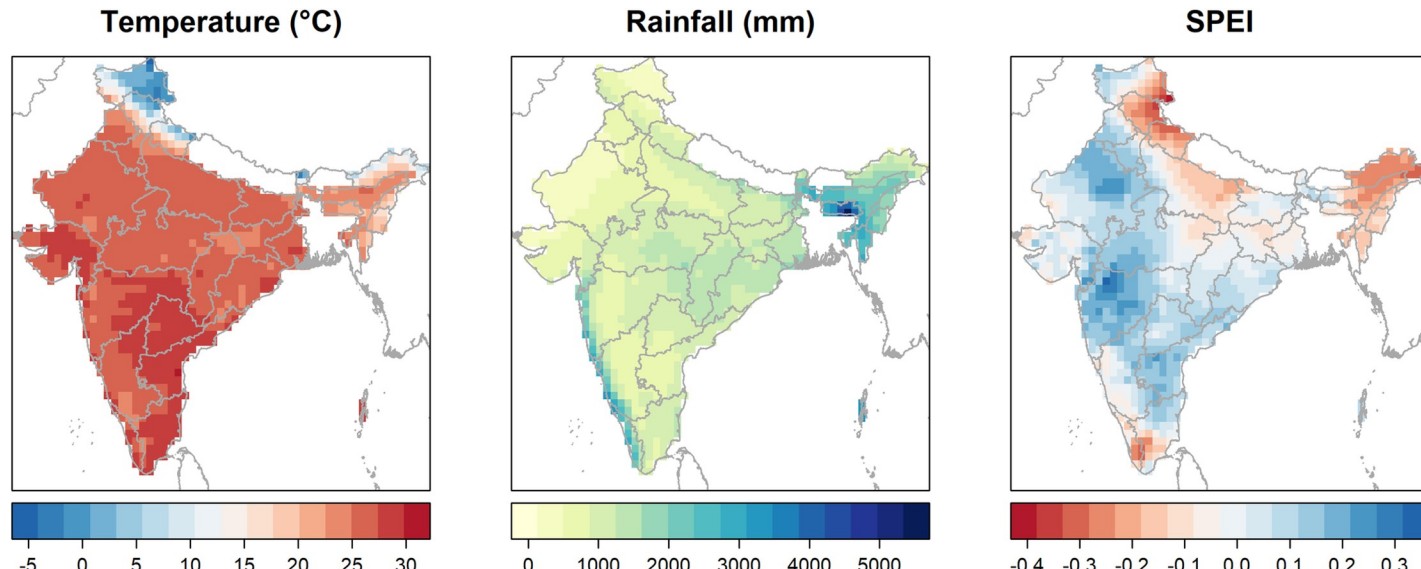

**Fig 3. Annual mean temperature (left), annual total rainfall (centre) and annual mean 1-month SPEI (right) at 0.5° spatial resolution, India, 2009–2016.** Map images were produced by the authors using climate data from the Climatic Research Unit at the University of East Anglia (CRU TS 3.25) [46] and administrative boundaries from the Global Administrative Areas Database (GADM). Both datasets are freely available for academic use.

which is the period of our analysis, SPEI varies from positive to negative in most climate zones, mainly due to changes in rainfall. However, no clear trends can be distinguished over this entire period.

## Estimation strategy

We use a multivariate logistic model to estimate the effect of SPEI on child health outcomes. The baseline model takes the following form:

$$y_{i,j} = \beta_1 \, SPEI_{k,t} + \delta \, X_i + f(a)_i + \mu_j + \epsilon_{i,j}$$

where $y_{i,j}$ is the health outcome of child $i$ in district $j$. $SPEI_{k,t}$ is the standardized precipitation and evapotranspiration index in grid cell $k$ during period $t$. To model the effects of weather shocks on stunting, we consider exposure to monsoon season SPEI in utero (while the child was in the womb) and during infancy (the first year of life). According to the fetal and infant origins hypothesis by Barker [50], [51], these early life periods are critical for children's physical development and can determine health outcomes in later life. For wasting and diarrhoea, we expect a more immediate response to climate shocks–for wasting we consider exposure to SPEI during the latest monsoon season before the child was measured, and for diarrhoea we consider exposure during the month of interview (see Fig 5).

$X_i$ is a column vector of control variables associated with childhood undernutrition. These are grouped into individual, maternal and household characteristics. Individual characteristics include child's gender, birth order, and whether the child was born a twin or not. We also include an interaction between child's gender and birth order. Maternal characteristics include age of the mother, height, exposure to mass media and highest educational level achieved. Household characteristics include wealth quintile, constructed using standard DHS procedures [52], gender of the household head, number of children under five years of age in the household, caste (scheduled caste, scheduled tribe, other backward caste or other caste) and religious affiliation (Hindu, Muslim, Christian or other religion). We also construct a dummy

**Fig 4.** Monthly temperature (A), rainfall (B), and 1-month SPEI (C) by climate zone in India (2009–2016 average). Mean monsoon season temperature (D), total monsoon season precipitation (E), and mean monsoon season SPEI (F) by climate zone in India for the period 2009–2016. Source: Climatic Research Unit at the University of East Anglia (CRU TS 3.25).

variable indicating whether the household has access to improved sanitation facility. According to the WHO/UNICEF definition [53], "improved" facilities include flush toilet, piped sewer system, septic tanks and other safe facilities which do not contaminate the living environment; "Unimproved" facilities include pit latrine, bucket toilet, other unsafe facilities or the general lack of sanitation facility on the premise. Descriptive statistics of the control variables are shown in S1 Table in S1 Appendix.

We also include fixed effects for month of birth and year of interview. $\mu_j$ are district fixed effects which capture policies and other factors at the district level which may influence childhood undernutrition. $f(a)_i$ is a restricted cubic age spline with knots at 6, 12, 18, 24, 36 and 48 months of age. The spline function fits polynomials of degree 3 between the defined knots in a

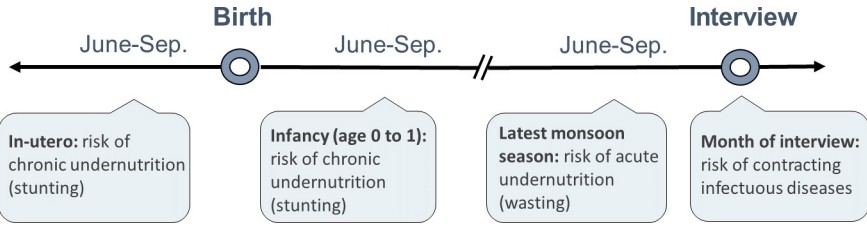

**Fig 5. Critical periods of exposure to climate shocks.**

way which ensures that levels and derivatives are equal on each side, and quadratic terms at each end. $\epsilon_{i,j}$ is the error term. Standard errors are clustered at the district level.

As discussed in the previous section, India spans a range of climatic zones. It can thus be expected that climate shocks will impact children's health differently depending on the distinct climate in their location. We expand our baseline model by the inclusion of an interaction term between SPEI and climate zone (Z) in order to detect spatial variations in climate-related vulnerabilities:

$$y_{i,j} = \beta_1 \, SPEI_{k,t} + \beta_2 (SPEI_{k,t} * Z_i) + \gamma Z_i + \delta \, X_i + f(a) + \mu_j + \epsilon_{i,j}$$

In addition to using SPEI as a continuous variable in our model, we construct variables for droughts and floods based on the SPEI categories described in Table 1 above. Monthly SPEI values below or equal to -1.5 are categorised as a drought events and values above or equal to 1.5 are categorized as flood events. We then assess the effect of experiencing at least one drought or flood event in a given monsoon season on the risk of undernutrition and diarrheal diseases.

It is important to note that high SPEI values cannot directly be interpreted as floods, as there are other factors contributing to flood developments besides high precipitation, such as soil saturation and river discharge [43], [45]. Nonetheless, heavy rainfall is the single most important factor contributing to flood build-ups.

## Results

### Descriptive analysis

Table 2 shows the sample descriptive statistics for the main variables of interest. The average HAZ score in the sample population is -1.42 and the average WHZ score is -0.95. Overall, 38% of children in our sample have stunted growth and 20% suffer from wasting. 16% of children

**Table 2. Descriptive statistics for main variables of interest.**

|  | Mean (Proportion) | Std. Dev. |
|---|---|---|
| HAZ | -1.42 | 1.78 |
| WHZ | -0.95 | 1.43 |
| stunted (HAZ<-2) | 0.38 | |
| wasted (WHZ<-2) | 0.20 | |
| severely stunted ((HAZ<-3) | 0.16 | |
| severely wasted (WHZ<-3) | 0.08 | |
| diarrhoea | 0.09 | |
| SPEI in utero | 0.04 | 0.58 |
| SPEI in infancy | 0.04 | 0.57 |
| SPEI latest monsoon season | -0.22 | 0.61 |
| SPEI month of interview | 0.41 | 1.00 |
| Drought in utero (SPEI≤-1.5) | 0.21 | |
| Flood in utero (SPEI≥1.5) | 0.30 | |
| Drought in infancy (SPEI≤-1.5) | 0.22 | |
| Flood in infancy (SPEI≥1.5) | 0.31 | |
| Drought latest monsoon season (SPEI≤-1.5) | 0.30 | |
| Flood latest monsoon season (SPEI≥1.5) | 0.16 | |

Summary statistics for the control variables used in the regression model are available in S1 Table in S1 Appendix.

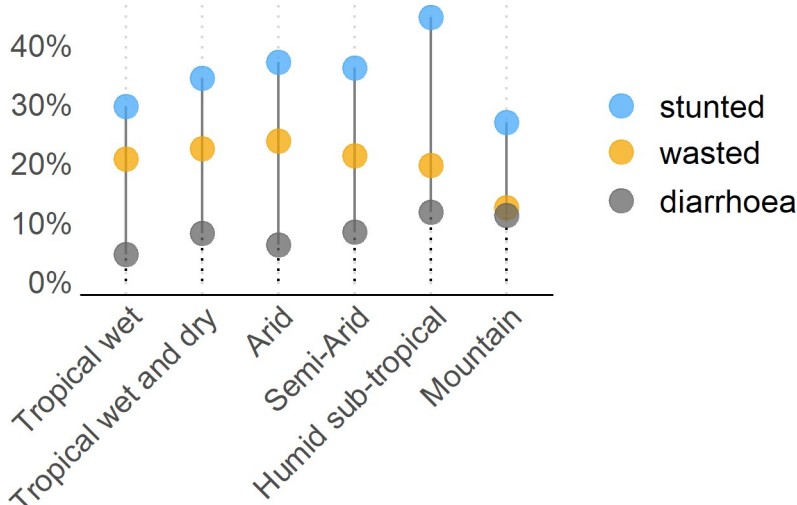

**Fig 6. Share of stunted and wasted children, and children with diarrhoea aged 0–5 by climate zone.** Source: Own calculations based on DHS India 2015–16. Notes: Sampling weights applied in the calculation of stunting, wasting and diarrhoeal prevalence. Climate zones based on Köppen-Geiger climate classification; see text for details.

are severely stunted and 8% are severely wasted. Fig 6 further shows the prevalence of under-nutrition across climate zones; Substantial differences can be observed. The highest prevalence of child stunting is seen in areas with humid sub-tropical climate (45% of children), whereas areas with arid, semi-arid, and tropical climate have a particularly high share of wasted children (over 20%). In contrast, the mountainous areas have the lowest prevalence of child stunting and wasting in the country but some of the highest prevalence of diarrhoea among children under the age of five (11%).

In terms of exposure to climate shocks, about 20% of children in our sample have experienced a monsoon season drought and 30% have experienced a monsoon season flood while in utero, according to our definition of floods and droughts. The shares of children exposed to monsoon season droughts and floods in infancy are similar to the above. 30% of children have been exposed to a drought and 16% to a flood in the latest monsoon season prior to the interview.

## Main results

Moving to the regression analysis, we find a strong association between monsoon rainfall variability and childhood undernutrition in India. Table 3, reports that an increase in the SPEI score of 1 during the in-utero period, which indicates wetter than usual monsoon season, reduces HAZ by 0.034 and increases the odds of stunting by 5% (both at 99% confidence, Table 3, col. 1 and 2). However, it is not clear whether this relationship is linear, it may be the case that both deficient and excessive precipitation during the monsoon season increases the risk of undernutrition. Looking at the effects of extreme climate events, we find evidence that the association is linear; Monsoon season droughts (SPEI$\leq$-1.5) reduce the risk of stunting by 5%, whereas floods (SPEI$\geq$1.5) increase it by 4% (both at 95% confidence, Table 3, col. 5). These effects are not trivial; Increasing mother's level of education from none to primary reduced the risk of stunting by a similar magnitude. The association between extreme climate events (droughts and floods) and HAZ follows the same direction but is weakly statistically significant. No association is found between monsoon rainfall shocks in utero and the risk of severe stunting.

**Table 3. Effects of monsoon season climate during in-utero on undernutrition, children aged 0–5.**

| | HAZ | stunted (HAZ<-2) | severely stunted (HAZ<-3) | HAZ | stunted (HAZ<-2) | severely stunted (HAZ<-3) |
|---|---|---|---|---|---|---|
| | OLS Coef. | Odds Ratio | Odds Ratio | OLS Coef. | Odds Ratio | Odds Ratio |
| SPEI in utero | -0.034** | 1.047** | 1.033 | | | |
| | [-0.057, -0.011] | [1.016, 1.080] | [0.993, 1.075] | | | |
| Drought in utero (SPEI≤-1.5) | | | | 0.030+ | 0.954* | 0.98 |
| | | | | [-0.005, 0.065] | [0.911, 0.998] | [0.925, 1.039] |
| Flood in utero (SPEI≥1.5) | | | | -0.027+ | 1.039+ | 1.029 |
| | | | | [-0.054, 0.001] | [1.001, 1.078] | [0.980, 1.080] |
| Controls | Yes | Yes | Yes | Yes | Yes | Yes |
| Age splines | Yes | Yes | Yes | Yes | Yes | Yes |
| District FEs | Yes | Yes | Yes | Yes | Yes | Yes |
| Obs. | 110,335 | 110,319 | 110,179 | 110,335 | 110,319 | 110,179 |
| (Pseudo) R$^2$ | 0.151 | 0.096 | 0.088 | 0.151 | 0.096 | 0.088 |

+<0.1

* <0.05

** <0.01

*** <0.001. 95% CIs are provided in parenthesis. Clustering at the district level. All control variables are included in the model but not displayed. Full results are available in S2 Table in S2 Appendix.

We find a strong association between monsoon rainfall variability in infancy and the risk of stunting and severe stunting. An increase in the SPEI score of 1 during infancy, which means wetter than usual climate, raises the odds of stunting by 4% and the odds of severe stunting by 7% (99% and 99.9% confidence, respectively, Table 4). Correspondingly, monsoon season droughts in infancy reduce the risk of stunting by and severe stunting while monsoon season floods increase it. The effect of floods on child stunting is comparable to living in a household without access to safe sanitation facility. We do not find an association between monsoon rainfall variability in infancy and HAZ.

Variations in SPEI during the latest monsoon season before the child was measured do not seem to affect WHZ scores and the risk of wasting and severe wasting (Table 5, col. 1–3). Looking at the effects of extreme climate events, however, we find some interesting associations. Both severely dry and severely wet monsoon seasons increase WHZ and reduce the risk of wasting (Table 5, col. 4 and 5). Severely dry monsoon weather (SPEI≤-1.5) reduces the odds of wasting by 11% (99.9% confidence), while severely wet weather (SPEI≥1.5) reduces the odds of severe wasting by 16% (95% confidence). These effects are equivalent to improving mother's level of education from none to post-secondary or moving households from the lowest wealth quintile to the second lowest.

Most of the control variables included in the model show significant effects (S2-S4 Tables in S2 Appendix). We discuss some of the more interesting associations bellow. Boys generally have lower HAZ and WHZ scores and are more likely to be stunted and wasted than girls. The female advantage in anthropometric status has been observed in a number of low- and middle-income countries [54], [55], and is usually attributed to biological [56] and behavioural [57] differences, however, the supporting evidence is very limited. Interestingly, a few studies on India find no gender differences in anthropometric status at young ages after accounting for household wealth, maternal education and other factors [58], [59]. What is more, the

**Table 4. Effects of monsoon season climate during infancy on undernutrition, children aged 0–5.**

| | HAZ | stunted (HAZ<-2) | severely stunted (HAZ<-3) | HAZ | stunted (HAZ<-2) | severely stunted (HAZ<-3) |
|---|---|---|---|---|---|---|
| | OLS Coef. | Odds Ratio | Odds Ratio | OLS Coef. | Odds Ratio | Odds Ratio |
| SPEI in infancy | -0.01 | 1.037** | 1.065*** | | | |
| | [-0.031, 0.010] | [1.011, 1.063] | [1.034, 1.096] | | | |
| Drought in infancy (SPEI≤-1.5) | | | | 0.017 | 0.966* | 0.945** |
| | | | | [-0.006, 0.040] | [0.935, 0.997] | [0.908, 0.984] |
| Flood in infancy (SPEI≥1.5) | | | | -0.002 | 1.040** | 1.042* |
| | | | | [-0.023, 0.019] | [1.011, 1.070] | [1.006, 1.080] |
| Controls | Yes | Yes | Yes | Yes | Yes | Yes |
| Age splines | Yes | Yes | Yes | Yes | Yes | Yes |
| District FEs | Yes | Yes | Yes | Yes | Yes | Yes |
| Obs. | 188,732 | 188,712 | 188,709 | 188,732 | 188,712 | 188,709 |
| (Pseudo) R² | 0.136 | 0.094 | 0.092 | 0.136 | 0.094 | 0.092 |

+<0.1

* <0.05

** <0.01

*** <0.001. 95% CIs are provided in parenthesis. Clustering at the district level. All control variables are included in the model but not displayed. Full results are available in S3 Table in S2 Appendix.

**Table 5. Effects of monsoon season climate prior to interview on undernutrition, children aged 0–5.**

| | WHZ | wasted (WHZ<-2) | severely wasted (WHZ<-3) | WHZ | wasted (WHZ<-2) | severely wasted (WHZ<-3) |
|---|---|---|---|---|---|---|
| | OLS Coef. | Odds Ratio | Odds Ratio | OLS Coef. | Odds Ratio | Odds Ratio |
| SPEI latest monsoon season | -0.023 | 1.036 | 0.928 | | | |
| | [-0.082, 0.036] | [0.942, 1.140] | [0.802, 1.075] | | | |
| Drought latest monsoon season (SPEI≤-1.5) | | | | 0.059* | 0.889*** | 0.918 |
| | | | | [0.013, 0.105] | [0.831, 0.950] | [0.822, 1.023] |
| Flood latest monsoon season (SPEI≥1.5) | | | | 0.062* | 0.910+ | 0.837* |
| | | | | [0.009, 0.114] | [0.827, 1.002] | [0.719, 0.975] |
| Controls | Yes | Yes | Yes | Yes | Yes | Yes |
| Age splines | Yes | Yes | Yes | Yes | Yes | Yes |
| District FEs | Yes | Yes | Yes | Yes | Yes | Yes |
| Obs. | 188,532 | 188,529 | 188,375 | 188,532 | 188,529 | 188,375 |
| (Pseudo) R² | 0.078 | 0.053 | 0.063 | 0.078 | 0.053 | 0.063 |

+<0.1

* <0.05

** <0.01

*** <0.001. 95% CIs are provided in parenthesis. Clustering at the district level. All control variables are included in the model but not displayed. Full results are available in S4 Table in S2 Appendix.

interaction term between child's sex and birth order in our model indicates that the male disadvantage in anthropometric status is reduced at higher birth order. This may reflect preferential treatment of boys who have older sisters, as some research suggests [60]. Gender discrimination practices which benefit male children are common in India, particularly among the Hindu population [60], [61].

We also find that the higher a mother's level of education is, the lower is the risk of a child being stunted or wasted; This is in line with previous research [55], [62], [63] and is usually attributed to better knowledge about nutrition, utilisation of antenatal and post-natal care, lower fertility (which means more resources available to fewer children), and higher autonomy in decision-making among educated women.

Expectedly, children born to wealthier households are less likely to be undernourished. Also, children living in households with access to safe sanitation facilities face lower risk of being stunted or wasted. In India, poor sanitation is a major health hazard; Over half of the population practices open defecation and lacks access to safe sanitation facilities [33], which has been linked to child stunting in earlier research [64].

Social class also matters; Children born in scheduled castes and tribes are more likely to be undernourished than those born in other, not socially disadvantaged, classes. In India, households belonging to scheduled groups are more susceptible to poverty [65], [66], which means higher risk of food insecurity, and reduced access to education and healthcare, all of which can negatively impact children's health. Such social and economic inequalities could exacerbate the effects of climate shocks and should be considered in future research.

A few robustness analyses were performed in order to eliminate potential biases in our sample composition and model specification. First, we exclude from our analysis children whose households have changed location between the time of exposure to the climate shock and the time of measurement. Second, an additional control variable for low birth weight is added to the model. We did not add this control variable in the main model since we expect that it would be correlated with climate conditions during the in-utero period. In both cases, the results remain robust.

As an additional robustness check, we construct an alternative climate measure. We use the monthly rainfall data from the CRU TS 3.25 database, which was used to generate the SPEI index, and restrict it to the monsoon months (June to September). We then generate a variable of rainfall anomalies as a standard deviation change in monsoon season rainfall from the location-specific long-term mean (1970–2016). We run the baseline model with new measure of rainfall anomalies. The results confirm that excessive precipitation during in-utero and infancy increases the risk of stunting, however, the effect sizes are reduced almost by half. This implies that chronic undernutrition is more sensitive to variations in SPEI rather than the rainfall-based measure. Additional information and results tables for all robustness checks are available in S3 Appendix.

## The role of infectious diseases

One of the mechanisms through which we expect climate shocks to affect child nutrition is the transmission of water-borne diseases, such as diarrheal infections. We investigate this mechanism by re-running the baseline model with diarrhoea as a dependent variable and climate in the month of interview as the main explanatory variable. The results presented in Table 6 show that indeed excessive precipitation in the month of interview is associated with an increased risk of contracting diarrhoea. We run the model again with an interaction term between the climate variable and the season of interview to test whether the latter moderates the effects of climate shocks on diarrhoea. Indeed, excessive rainfall during the monsoon

**Table 6. Effects of climate in the month of interview on the risk of diarrhoea, children aged 0–5.**

| | diarrhoea | diarrhoea |
|---|---|---|
| | **Odds Ratio** | **Odds Ratio** |
| SPEI month of interview | 1.066* | |
| | [1.011, 1.124] | |
| SPEI month of interview: Winter season | | 1.08 |
| | | [0.976, 1.195] |
| SPEI month of interview: Summer season | | 1.016 |
| | | [0.941, 1.097] |
| SPEI month of interview: Monsoon season | | 1.159** |
| | | [1.051, 1.279] |
| SPEI month of interview: Post-monsoon season | | 1.193 |
| | | [0.958, 1.486] |
| Controls | Yes | Yes |
| Age splines | Yes | Yes |
| District FEs | Yes | Yes |
| Obs. | 225,577 | 188,642 |
| Pseudo R$^2$ | 0.086 | 0.086 |

+ $<$0.1

* $<$0.05

** $<$0.01

*** $<$0.001. SPEI is calculated on a 3-month scale. 95% CIs are provided in parenthesis. Clustering at the district level. All control variables are included in the model but not displayed. Full results are available in S5 Table in S2 Appendix.

months is associated with an increased risk of contracting diarrhoea, while no significant effects are found in the other seasons (Table 6, col.2).

To further explore the role of infectious diseases in mediating the effect of excessive rainfall on child undernutrition, we expand the baseline model with an interaction term between SPEI and a variable indicating the type of sanitation facility available to the household. It is expected that the incidence of water-borne diseases will increase in a wet environment which is already contaminated, i.e. among households who lack access to appropriate sanitation facility. The results suggest that this may be the case. Among those household who lack access to improved sanitation facility, children face an increased risk of stunting due to excessive rainfall (Table 7). No effects are found among households who have access to improved sanitation facility.

## Stunting and "catch-up" growth

Some evidence suggests that children who have been stunted early in life can become "late bloomers", meaning that they are able to catch up with their peers and reach their full growth potential at a later age [67], [68]. We assess the possibility of such "catch-up" growth by running the analysis on sub-samples of older children. The results presented in Table 8 indeed provide some evidence in support of this. Above age three, excessive monsoon rainfall during infancy is no longer associated with child stunting. Similarly, above age one excessive monsoon rainfall in utero does not seem to affect the risk of stunting. However, these results might also be driven by selective survival, meaning that children who have been most affected by climate shocks in utero or during infancy are less likely to survive to later ages and would drop out from our sample. Further research is needed to determine the possibility of "catch-up" growth and selective survival, preferably with panel data.

**Table 7. Effects of monsoon season SPEI on undernutrition by access to improved sanitation facility, children aged 0–5.**

| | stunted | stunted | wasted |
|---|---|---|---|
| | **Odds ratio** | **Odds ratio** | **Odds ratio** |
| SPEI in utero: Unimproved sanitation facility | 1.060*** | | |
| | [1.024,1.097] | | |
| SPEI in utero: Improved sanitation facility | 1.022 | | |
| | [0.975,1.071] | | |
| SPEI in infancy: Unimproved sanitation facility | | 1.054*** | |
| | | [1.024,1.085] | |
| SPEI in infancy: Improved sanitation facility | | 1.003 | |
| | | [0.967,1.040] | |
| SPEI latest monsoon season: Unimproved sanitation facility | | | 1.039 |
| | | | [0.944,1.144] |
| SPEI latest monsoon season: Improved sanitation facility | | | 1.032 |
| | | | [0.931,1.144] |
| Controls | Yes | Yes | Yes |
| Age splines | Yes | Yes | Yes |
| District FEs | Yes | Yes | Yes |
| Obs. | 110,319 | 188,712 | 188,529 |
| Pseudo R2 | 0.096 | 0.094 | 0.053 |

+<0.1

* <0.05

** <0.01

*** <0.001. 95% CIs are provided in parenthesis. Clustering at the district level. All control variables are included in the model but not displayed. Full results are available in S6 Table in S2 Appendix.

## Differences by climate zone

In this section, we explore spatial differences in climate-related vulnerabilities in India by including an interaction term in our baseline model between the SPEI variable and a

**Table 8. Effects of monsoon season SPEI on the risk of stunting by age at measurement.**

| | **Age > 1** | **Age > 2** | **Age > 3** | **Age > 4** |
|---|---|---|---|---|
| | **Odds Ratio** | **Odds Ratio** | **Odds Ratio** | **Odds Ratio** |
| SPEI in utero | 1.011 | 0.998 | 0.989 | 1.031 |
| | [0.979, 1.043] | [0.964, 1.034] | [0.950, 1.029] | [0.918, 1.157] |
| Obs. | 88,783 | 66,565 | 44,623 | 21,667 |
| Pseudo R2 | 0.091 | 0.101 | 0.107 | 0.116 |
| SPEI in infancy | 1.034** | 1.029+ | 1.029 | 1.024 |
| | [1.008, 1.060] | [0.999, 1.060] | [0.992, 1.068] | [0.974, 1.075] |
| Obs. | 177,273 | 133,894 | 90,126 | 44,127 |
| Pseudo $R^2$ | 0.089 | 0.097 | 0.099 | 0.102 |

+<0.1

* <0.05

** <0.01

*** <0.001. 95% CIs are provided in parenthesis. Clustering at the district level. All control variables are included in the model but not displayed. Full results are available in S7 and S8 Tables in S2 Appendix.

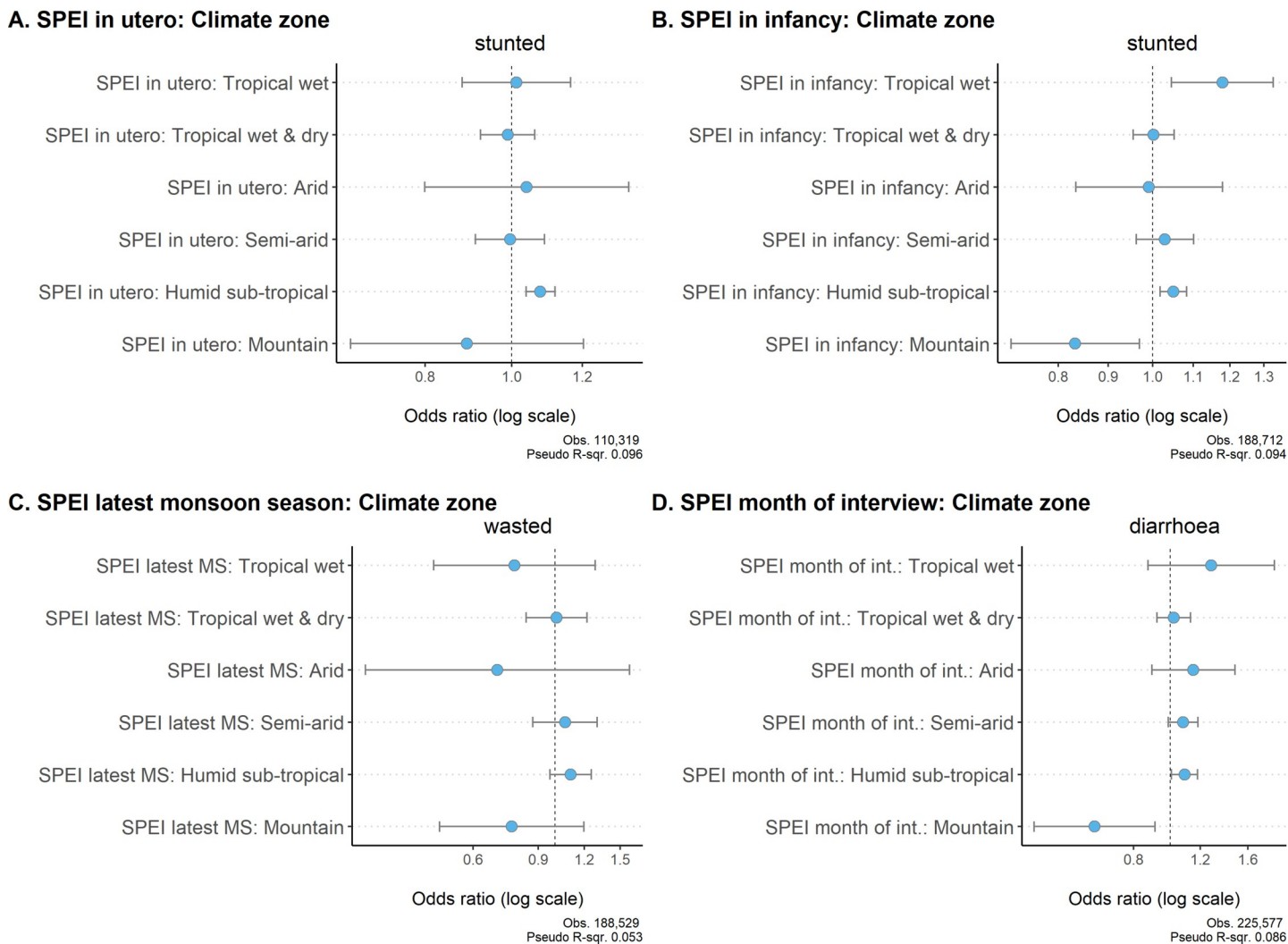

**Fig 7. Effects of SPEI on undernutrition by climate zone, children aged 0–59 months.** Figure shows odds ratios with 95% CIs. All control variables are included in the model but not displayed. Results are available in S9 Table in S2 Appendix.

categorical variable denoting the six main climate zones in the country. The results consistently show that children living in the humid sub-tropical climate face higher risk of being stunted and contracting diarrhoeal infections due to excessive rainfall; The odds of stunting are 8% higher (99.9% confidence) due to exposure to excessive monsoon rainfall in utero, and 5% higher (99% confidence) due to exposure to excessive monsoon rainfall during infancy (Fig 7A and 7B). The risk of contracting diarrhoea is 9% higher if the month of interview was abnormally wet (Fig 7D). Excessive monsoon rainfall seems to increase the risk of wasting as well in the humid sub-tropical climate, however, the effect is not statistically significant (Fig 7D).

In the tropical wet climate, a positive association is also found between excessive monsoon rainfall during infancy and stunting, with an increase in the SPEI score of 1 associated with an 18% increase in the risk of stunting (99% confidence). In contrast, in the mountainous regions in north India excessive monsoon rainfall during infancy is associated with a reduced risk of stunting and abnormally wet month of interview is associated with a reduced risk of contracting diarrhoea (Fig 7).

## Discussion and conclusions

We show that exposure to excessive monsoon rainfall in utero and during the first year after birth is associated with an increased risk of undernutrition among children under the age of five. Children who have experienced excessive monsoon rainfall in utero are more likely to be too short for their age and to be stunted. Children who were exposed to similar rainfall shocks in infancy are more likely to be stunted and severely stunted. These findings are in line with previous research, which shows that the period during pregnancy and the first 12 months of life are critical for children's physical development [69]. The effects of early-life rainfall shocks on stunting seem to reduce as children get older, which could be due to "catch-up" growth or selective survival; Further research is needed to understand this phenomenon.

We also find that abnormally dry and abnormally wet monsoon seasons are likewise associated with reduced risk of wasting and higher weight-for-age z-scores. This implies that the relation between excessive precipitation and child undernutrition is not one-directional but could be modified by contextual factor. It should be noted that our analysis of wasting is limited to climate shocks experienced during the most recent monsoon season before the interview. Since we only use one DHS wave, all interviews have taken place between 2015 and 2016, which implies limited variability in climate conditions. Adding more DHS waves can help better understand the short-term effects of monsoon rainfall shocks on wasting.

The transmission of infectious diseases is a main channel through which heavy rainfall can affect child nutrition. Considering that 71% of rural households in India do not treat water prior to drinking and 54% have no access to toilet facilities [34], severe monsoon rainfall could add to the risk of contracting diarrhoeal and other diseases transmitted through water. Such a spike in communicable diseases was observed in Kerala in the aftermath of the 2018 floods. Studies show that child nutrition and early development can be affected by the presence of such infections [70–72].

We assess the link between rainfall variability and diarrhoeal infections among the sample of children aged under five. Indeed, the results indicate that excessive precipitation, particularly during the monsoon months, is associated with an increased risk of contracting diarrhoea. We additionally observed that children who live in households without access to safe sanitation facilities face increased risk of stunting due to heavy monsoon rainfall. We do not find such risk for children living in households with access to appropriate sanitation facilities. These findings confirm the role of infectious diseases in mediating the effect of rainfall shocks on child health.

Disaggregating the analysis by climate zone in India, we show that there are large spatial variations in climate-related vulnerabilities. Excessive monsoon precipitation increases the risk of stunting and diarrheal infections for children living in tropical wet and humid sub-tropical climate zones. In contrast, the risk of stunting is reduced in the mountainous regions in northern India. We do not find evidence that monsoon rainfall variability affects child undernutrition in other climate zones.

The tropical wet and humid sub-tropical zones receive some of the highest levels of monsoon rainfall in normal times, which may explain the positive association between excessive precipitation and childhood undernutrition in these areas. Monsoon rainfall above the norm in these zones is likely to contaminate water and damage crops [73], and in the extreme case, trigger floods. In contrast, the mountainous region in northern India is relatively dry; Increased precipitation during the monsoon season implies improved access to clean water, as well as increased crop yield [73], both of which can benefit child health. Deficient water in the summer months has become a particularly pressing problem in the Indo- Gangetic plain; Decreasing snowcap is leading to a reduced water supply in the summer season and affecting poorly irrigated farmers [74]. This may explain the reduction in diarrheal infections and child stunting during periods of high precipitation in the mountainous regions.

When it comes to identifying climate-related vulnerabilities, there is a need to better understand the role of contextual factors, such as climatic and agroecological factors and underlying inequalities. Future research should focus on identifying "hotspots" where climate change impacts will be most pronounced.

This study has certain limitations which should also be considered. We are not able to establish a causal link between monsoon rainfall variability, infectious diseases and child undernutrition with the available cross-sectional data. Moreover, our estimates may be biased due to selective survival. If this is the case, we are likely to underestimate the effects of climate shocks on child undernutrition. Our analysis of diarrheal infections relies on self-reported data, which is subject to misreporting. In addition, repeated exposure to climate shocks is likely to pose an even greater risk to child health, however, we only consider single periods of exposure. In addition to addressing the above issues, new research should consider alternative measures of climate shocks, such as delays in monsoon precipitation and duration of the rainy season. For this, using daily climatological data would be more appropriate.

In conclusion, the intensity and frequency extreme climate events is projected to increase in South Asia over the coming decades [4], [15]. Therefore, there is an urgent need to better understand the potential hazards for children and create early-response mechanisms. The present study stresses the importance of increasing assistance to communities in flood-prone areas. Interventions should be focused on households with pregnant women and infants. Immunization, improved access to healthcare, and the provision of safe drinking water and sanitation facilities are some of the measures which can be taken to improve the prospects of children in flood-prone areas in India.

## Supporting information

**S1 Appendix. Data.**
(DOCX)

**S2 Appendix. Results tables.**
(DOCX)

**S3 Appendix. Robustness analysis.**
(DOCX)

## Author Contributions

**Conceptualization:** Anna Dimitrova, Jayanta Kumar Bora.

**Data curation:** Anna Dimitrova.

**Formal analysis:** Anna Dimitrova.

**Investigation:** Anna Dimitrova.

**Methodology:** Anna Dimitrova.

**Visualization:** Anna Dimitrova.

**Writing – original draft:** Anna Dimitrova, Jayanta Kumar Bora.

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
