## [Decision Letter · Decision Letter 0]

21 Jan 2020

PONE-D-19-29467

Monsoon weather and early childhood health in India

PLOS ONE

Dear Ms. Dimitrova,

Thank you for submitting your manuscript to PLOS ONE. After careful consideration, we feel that it has merit but does not fully meet PLOS ONE’s publication criteria as it currently stands. Therefore, we invite you to submit a revised version of the manuscript that addresses the points raised during the review process.

We would appreciate receiving your revised manuscript by Mar 06 2020 11:59PM. To enhance the reproducibility of your results, we recommend that if applicable you deposit your laboratory protocols in protocols.io, where a protocol can be assigned its own identifier (DOI) such that it can be cited independently in the future. For instructions see: http://journals.plos.org/plosone/s/submission-guidelines#loc-laboratory-protocols

We look forward to receiving your revised manuscript.

Kind regards,

Kannan Navaneetham

Academic Editor

PLOS ONE

Journal Requirements:

2. Please be wary of including causal statements in this manuscript, including in the Abstract.

3. We note that Figures #2 and S1 in your submission contain map images which may be copyrighted. All PLOS content is published under the Creative Commons Attribution License (CC BY 4.0), which means that the manuscript, images, and Supporting Information files will be freely available online, and any third party is permitted to access, download, copy, distribute, and use these materials in any way, even commercially, with proper attribution. For these reasons, we cannot publish previously copyrighted maps or satellite images created using proprietary data, such as Google software (Google Maps, Street View, and Earth). For more information, see our copyright guidelines: http://journals.plos.org/plosone/s/licenses-and-copyright.

a.    You may seek permission from the original copyright holder of Figures #2 and S1 to publish the content specifically under the CC BY 4.0 license. 

Reviewers' comments:

Reviewer's Responses to Questions

**Comments to the Author**

1. Is the manuscript technically sound, and do the data support the conclusions?

Reviewer #1: Partly

Reviewer #2: Yes

Reviewer #3: Yes

2. Has the statistical analysis been performed appropriately and rigorously? 

Reviewer #1: No

Reviewer #2: Yes

Reviewer #3: Yes

3. Have the authors made all data underlying the findings in their manuscript fully available?

Reviewer #1: Yes

Reviewer #2: Yes

Reviewer #3: Yes

4. Is the manuscript presented in an intelligible fashion and written in standard English?

Reviewer #1: No

Reviewer #2: Yes

Reviewer #3: Yes

5. Review Comments to the Author

Reviewer #1: The purpose of the paper is to assess the effect of monsoon weather on early childhood health. Similar exercises have been conducted in India and other countries. This paper is too ambitious, given the kind of data that is available. It rightly notes that causality issues cannot be addressed with cross-section data. It would be much more realistic to try and link weather conditions with reported morbidity of children, on which there is data in DHS.

There is a huge body of literature on child undernutrition in India. The paper inadequately summarizes the literature. It focuses more on the physical geography aspect of undernutrition. In constructing a household model to explain child undernutrition it does not control for child's birth-order, its interaction with child's sex, mother's nutritional status, access to sanitation facilities and receiving benefits from a public programme (namely the ICDS). These are the interesting and debated areas in child undernutrition in India (see the works of Dean Spears, Rohini Somanathan, Seema Jayachandran, etc.).

The manuscript also needs major revisions in terms of language editing.

Reviewer #2: Thank you for this interesting and useful contribution to the literature. Not enough work on climate and nutrition has been focused outside of Africa, so it is good to see some work from south Asia. Additionally, it is great to see work examining how prevailing climate regimes affect vulnerability. Nevertheless, there are some things that could still be improved about the paper.

Abstract - "implications on" should be "impacts on" or "implications for"

62 - "extent"

76-87 - I would here include papers that also look at food security metrics like HFIAS and HHS in addution to nutrition outcomes in relation to flooding in south asia, such as Cooper 2019 Population and Environment

198 - "overlaid"

219-226 - What about birth order? This is commonly included in regressions

263 - why are you interpreting SPEI in standard deviations? I would say "an increase in the SPEI score of 1 would like to an XX% increase in the probability of..."

342 - Maybe specifically mention that this could be address with DHS variables like the incidence of diarrheal disease as well as dietary diversity.

356 - I would cite here the Eissler 2019 Global Environmental Change paper

335-341, be more explicit about why tropical wet and humid sub-tropical are most prone to monsoon floodings. If we are defining flooding by SPEI, then a high SPEI value could could happen anywhere. So maybe say somehting more like "in areas that are already more wet, positive rainfall anomalies are more damaging"

Broader modeling issues

- Why do you use logitistic regression with a binary outcome variabe for stunting, wasting and underweight? Why not just model the Z-score directy in a normal regression? The cut-off of -2 for a Z-score is arbitrary and is not meaningful for diagnosing stunting in indviduals (See Perumal, 2018, Journal of Nutrition). Furthermore, you are losing a lot of information in your data by discretizing a continuous variable. I recommend either justifying the use of logisitic regression with a -2 Z-score cutoff or re-running the analysis with continous outcome variables.

- I really like how you modeled stunting and wasting based on climate shocks at different periods of children's lives, that approach is well-grounded based on our theoretical expectations. But why include WAZ/underweight at all? This is a less informative indicator and its inclusion is not well-justified. I would cut it.

- I like that you have quarterly fixed effects in the analysis, but I would include monthly fixed effects, or at least a discussion of the Larsen, 2019, Demography paper on how mis-reporting the month of birth can cause mis-estimation of HAZ scores.

- While your results definitely jibe with the idea that shocks in early life affect stunting later in life, you should address the well-established phenomenon of catch-up growth. I would like to see another model assessing whether the effects of early-life flooding on stunting is different for younger children versus older children. If this effect is small for older children, this would indicate that some catch-up growth has occured. However, if that is not possible, the phenomenon of catch-up growth should at least be discussed in the limitations paragraph.

Overall Discussion Issues

- Why did you see no effect for urban children? It is understandable why drought wouldnt matter -they are not producing their own food. But presumeably the effects of flooding on sanitation and infectious disease would matter in urban areas? Maybe use the DHS to add summary statistics about whether urban households have improved sanitation facilities, if the India DHS has that data.

- Also, address why you saw an opposite effect (floods are good for nutrition) in mountains?

- I would add the caveat that floods are hard to detect with 9-month or 12-month SPEI. A high 12-month SPEI could indicate several mildly wet months, or one severe flood followed by eleven normal months, and while these twe scenarios will have the same SPEI score, they could have very divergent effects on nutrition. I would like to see another analysis that addresses this, perhaps by using the count of months with 1-month SPEI greater than a threshhold. However, if this is not feasible it should be at least be discussed in the caveats section.

- You mention the issue of migration - most DHS surveys have data on how long a household has been in at their current location, so you can exclude children that were not at that location when they were in utero or in their first year. You should definitely address this, or at least mention that the India DHS does not include this variable.

- Interpret not just the sign and significant but also the magnitude of SPEI changes. How does the increase in the odds of stunting associated with and SPEI of 2 compare with the increase in the odds of stunting associated with having a less educated mother or being born a twin? This is important to contextualize the potential impact of climate shocks.

Reviewer #3: PONE-D-19-29467

Monsoon weather and early childhood health in India

The paper examines the association between exposure to extreme weather shocks (either drought or flood) during in-utero or infancy period (age 1) on childhood undernutrition (measured by stunting, wasting, and underweight). The paper finds that an increase in monsoon season SPEI (standardized precipitation and evapotranspiration index over the monsoon season) during the monsoon months is associated with higher prevalence of childhood stunting in India. The effect was stronger in rural areas compared with urban areas. Additionally, neither wasting nor probability of being underweight was affected by SPEI.

This is a worthwhile contribution on an important topic; however, I have a few concerns with the way the paper is put together in this version, and would suggest attending to these issues:

1. There are several papers that measure extreme weather shocks by amount of rainfall. The deviation of current year/season rainfall amount from the last 25-30 years average is used as proxy measure of droughts/floods. This paper uses another measure SPEI. It would be useful to have some discussion in the paper on comparative advantage of SPEI over rainfall-based definition of droughts/floods. Is SPEI better measure than rainfall shocks?

2. Related to previous comments, policymakers identify districts based on amount of deficient rainfall and then roll out public polices to moderate the effects of negative rainfall. So from policy perspective, rainfall shocks seem more intuitive than SPEI. Therefore, I would like to authors to include discussion on the difference between SPEI/rainfall shocks and comment which measure is more useful for policy design.

3. In the current version, SPEI has been used as continuous variable. The authors motivated the paper by explaining the negative effects of droughts/floods on childhood outcomes. In the current analysis, 1 S.D. increase in SPEI does not mean whether it is a drought or flood. In that vein, I am suggesting if authors could conduct an additional analysis and create a binary indicator of floods/droughts using the SPEI index. For example, club SPEI ≤ -2 Extreme drought and -2 < SPEI ≤ -1.5 Severe drought as binary indicator of drought and similarly for flood.

4. Do climate regimes map to SPEI ranges in Table 1?

5. Why maternal age at birth is included as control and not the current age of mothers? Provide an explanation.

6. I would like the baseline specification to include some indicator of birth outcomes as control covariate, for example, I would suggest to include birthweight as covariate in the multivariate logistic model.

7. Why age splines are included and not age in continuous months/years?

8. Access to sanitation in year 1 has been found to affect stunting, could authors include toilet access in the control set of variables?

9. Line 262, results interpretation; “reports that 1 standard deviation (SD) increase in monsoon season SPEI in utero……”. Could we interpret this as “increase in drought…”? or is it possible to map this SPEI increase to drought? I am not familiar with SPEI literature so clarification on this would help the readers.

10. Line 350- Authors discuss several hypothesis about mechanism. One of the mechanisms could be incidence of diarrhea. The DHS survey contains information on diarrhea for the sample children. I would like authors to explore this a bit and run the baseline specification with diarrhea as an outcome.

6. PLOS authors have the option to publish the peer review history of their article (what does this mean?). If published, this will include your full peer review and any attached files.

Reviewer #1: Yes: Simantini Mukhopadhyay

Reviewer #2: Yes: Matthew Cooper

Reviewer #3: No

---

## [Author Response · Author response to Decision Letter 0]

4 Mar 2020

JOURNAL REQUIREMENTS:

Response: Thank you for pointing to us this issue. We have made sure to meet PLOS ONE’s style requirements.

2. Please be wary of including causal statements in this manuscript, including in the Abstract.

Response: Thank you for this comment, we have ensured that no causal statements remain in the manuscript.

3. We note that Figures #2 and S1 in your submission contain map images which may be copyrighted. 

Response: The maps shown in the manuscript were produced by the authors based on freely available data. We have made sure that we comply with all condition when reproducing the map images and added disclaimers when necessary. See for example line 222: 

“The data used for producing this image is freely available under the Creative Commons license (accessed on 21.Feb.2020).”

COMMENTS BY REVIEWER 1

4. The purpose of the paper is to assess the effect of monsoon weather on early childhood health. Similar exercises have been conducted in India and other countries. This paper is too ambitious, given the kind of data that is available. It rightly notes that causality issues cannot be addressed with cross-section data. It would be much more realistic to try and link weather conditions with reported morbidity of children, on which there is data in DHS.

Response: It is true that previous studies have investigated the effect of monsoon weather shocks on child health in India, however, most of these studies rely on small samples sizes and/or are focused on specific locations (see our summary of the literature on lines 71-96). To the best of our knowledge, no previous study has used the SPEI index to measure climate shocks in this context. We believe that out paper makes a novel contribution to the literature by utilising a nationally representative survey data, constructing measures of both deficient and excessive monsoon rainfall and pointing out differences in vulnerability by climate zones in the country. 

While it is not possible to infer causality with the available data, we have managed to establish a link between excessive precipitation and child undernutrition. Considering that the climate shocks are essentially exogenous events, it is unlikely that the effects are driven by other factors. The analysis of diarrhoea, which we added, points that this relationship runs through increased incidence of infectious diseases. We also looked at differences in climate-related vulnerabilities by type of sanitation facility available to the household. The results also give support to the climate-diseases-nutrition pathway. 

5. There is a huge body of literature on child undernutrition in India. The paper inadequately summarizes the literature. It focuses more on the physical geography aspect of undernutrition. In constructing a household model to explain child undernutrition it does not control for child's birth-order, its interaction with child's sex, mother's nutritional status, access to sanitation facilities and receiving benefits from a public programme (namely the ICDS). These are the interesting and debated areas in child undernutrition in India (see the works of Dean Spears, Rohini Somanathan, Seema Jayachandran, etc.). 

Response: The main focus of the paper is indeed on the effects of climate shocks on child undernutrition and the way these are moderated by geographical location. There is already a large body of literature which has identified important individual and household-level factors that affect child nutrition. Our goal is to go beyond the household-level model and to better understand the risks associated with climate shocks. We consider this to be a timely research topic given that climate change is already shifting rainfall patterns in India and the incidence of extreme climate events is increasing. It is still unclear how human health will be affected.

Following the reviewer’s recommendation, we have included birth order, the interaction between child sex and birth order, and sanitation facility as control variables in the model. These indeed appear to be important predictors of child health status. We have not included mother’s nutrition status as a control variable since we expect that it would act as a mediator in the relationship between climate shocks and child health. For example, women who have been exposed to a climate shocks during pregnancy would be more likely to be malnourished and to give birth to a malnourished child. 

Following the reviewer’s recommendation, we have extended the analysis of some key control variables included in the model, such as child’s gender, maternal education, family’s caste and wealth. See lines 364-391, for example:

“Most of the control variables included in the model show significant effects (S2-S4 Tables in S2 Appendix). We discuss some of the more interesting associations bellow. Boys generally have lower HAZ and WHZ scores and are more likely to be stunted and wasted than girls. The female advantage in anthropometric status has been observed in a number of low- and middle-income countries [52],[53], and is usually attributed to biological [54] and behavioural [55] differences, however, the supporting evidence is very limited. Interestingly, few studies on India find no gender differences in anthropometric status at young ages after accounting for household wealth, maternal education and other factors [56],[57]. What is more, the interaction term between child’s sex and birth order in our model indicates that the male disadvantage in anthropometric status is reduced at higher birth order. This may reflect preferential treatment of boys who have older sisters, as some research suggests [58]. Gender discrimination practices which benefit male children are common in India, particularly among the Hindu population [58],[59]…”

We have also referenced research recommended by the reviewer, for example by Dean Spears and Seema Jayachandran.

Concerning the benefits received from public programmes (such as the ICDS), we have decided not to include them in the analysis. We suspect that distribution of such benefits would correlate with maternal and household characteristics already controlled for in the model, such as wealth and education. Analysing the effectiveness of such programmes goes beyond the scope of this paper. Nonetheless, the impact of such programmes on child health is of high policy relevance and we believe that a separate study should be dedicated to investigating this further.

6. The manuscript also needs major revisions in terms of language editing.

Response: The manuscript was proofread by a native speaker in order to resolve language issues.

COMMENTS BY REVIEWER 2

Comments

7. Abstract - "implications on" should be "impacts on" or "implications for"

Response: Thank you for detecting this mistake. We have changed the text to “impacts on”.

8. 62 - "extent"

Response: Thank you for detecting the typo. We have corrected the text.

9. 76-87 - I would here include papers that also look at food security metrics like HFIAS and HHS in addition to nutrition outcomes in relation to flooding in south asia, such as Cooper 2019 Population and Environment

Response: Thank you for pointing us to this literature, we have included a reference to the Cooper 2019 paper. See lines 91-94: 

“... Another study used the Household Hunger Scale (HHS), which is a household food security metric, in addition to child anthropometric status to detect climate-induced hunger in Bangladesh and Ghana [30]. The study identified a link between increased rainfall and hunger in Bangladesh….” 

10. 198 - "overlaid"

Response: Thank you for detecting this typo. We have corrected the text.

11. 219-226 - What about birth order? This is commonly included in regressions

Response: Following the reviewer’s suggestion, we have included birth order as a control variable in the regression model. It does indeed appear to be an important predictor of child undernutrition.

12. 263 - why are you interpreting SPEI in standard deviations? I would say "an increase in the SPEI score of 1 would like to an XX% increase in the probability of..."

Response: We changed the interpretation of SPEI scores as suggested throughout the text.

13. 342 - Maybe specifically mention that this could be address with DHS variables like the incidence of diarrheal disease as well as dietary diversity.

Response: Following a suggestion by Reviewer 3, we have included diarrhoea in our analysis. In particular, we have investigated the effects of experiencing climate shocks during the month of interview on the risk of contracting diarrhoea in the two weeks preceding the interview. We have indeed found a strong association between excessive precipitation in the monsoon months and the risk of contracting diarrhoea, see Table 6 and Fig. 5, and the corresponding text. 

14. 356 - I would cite here the Eissler 2019 Global Environmental Change paper

Response: Could you specify which paper? We have failed to identify the paper.

15. 335-341, be more explicit about why tropical wet and humid sub-tropical are most prone to monsoon floodings. If we are defining flooding by SPEI, then a high SPEI value could happen anywhere. So maybe say somehting more like "in areas that are already more wet, positive rainfall anomalies are more damaging"

Response: Thank you for this suggestion. We have included the following text in the discussion section (lines 506-509):

“The tropical wet and humid sub-tropical zones receive some of the highest levels of monsoon rainfall in normal times, which may explain the positive association between excessive precipitation and childhood undernutrition in these areas. Monsoon rainfall above the norm in these zones is likely to contaminate water and damage crops [73], and in the extreme case, trigger floods...”

Broader modeling issues

16. Why do you use logitistic regression with a binary outcome variabe for stunting, wasting and underweight? Why not just model the Z-score directy in a normal regression? The cut-off of -2 for a Z-score is arbitrary and is not meaningful for diagnosing stunting in indviduals (See Perumal, 2018, Journal of Nutrition). Furthermore, you are losing a lot of information in your data by discretizing a continuous variable. I recommend either justifying the use of logisitic regression with a -2 Z-score cutoff or re-running the analysis with continous outcome variables.

Response: Following the reviewer’s suggestion, we have added analysis of HAZ and WHZ scores as continuous variables using an OLS regression, see Tables 3 to 5 and the corresponding text. We have also added analysis of severe stunting (HAZ<-3) and severe wasting (WHZ<-3) to complement the previous findings. We agree that using arbitrary cut-offs is not ideal for diagnosing undernutrition in individuals. However, a high prevalence of stunting in a given population is indicative of poor environmental and socioeconomic conditions. Increasing levels of stunting and severe stunting are alarming, particularly in populations which already have a high burden of undernutrition. Additionally, the reference categories are widely used in the literature, especially for comparison and monitoring purposes, and are easily communicated to policymakers. For these reasons, we decided to keep the analysis of stunting and wasting while complementing it with HAZ and WHZ scores. We added a discussion of the above issues in the text, see lines 166-173:

“The usefulness of reference values, such as HAZ < -2, for detecting linear growth deficits in children has been a subject of debate. At the individual level, applying an arbitrary cut-off point can lead to underestimation of health problems; Whether a child is slightly below or slightly above the cut-off line should not make big a difference to their health condition [38]. On a societal level, however, such cut-off points are informative as they allow the monitoring of child malnutrition over time [39], and reveal important socioeconomic inequalities when calculated for different population sub-groups [40],[41]. For the purpose of this analysis, HAZ and WHZ scores are used both as continuous variables and dichotomised using different cut-off points as described above.”

17. I really like how you modeled stunting and wasting based on climate shocks at different periods of children's lives, that approach is well-grounded based on our theoretical expectations. But why include WAZ/underweight at all? This is a less informative indicator and its inclusion is not well-justified. I would cut it.

Response: We have removed WAZ/underweight from the analysis.

18. I like that you have quarterly fixed effects in the analysis, but I would include monthly fixed effects, or at least a discussion of the Larsen, 2019, Demography paper on how mis-reporting the month of birth can cause mis-estimation of HAZ scores.

Response: We have included month of birth fixed effects in the analysis.

19. While your results definitely jibe with the idea that shocks in early life affect stunting later in life, you should address the well-established phenomenon of catch-up growth. I would like to see another model assessing whether the effects of early-life flooding on stunting is different for younger children versus older children. If this effect is small for older children, this would indicate that some catch-up growth has occured. However, if that is not possible, the phenomenon of catch-up growth should at least be discussed in the limitations paragraph.

Response: Thank you for this interesting idea. In order to assess the possibility of catch-up growth, we have run the analysis separately for children of different age-groups. The results indeed show that the effects of climate shocks on stunting are reduced beyond age 2 and disappear beyond age 3. See Table 8 and the corresponding text (lines 437-446):

“Some evidence suggests that children who have been stunted early in life can become “late bloomers”, meaning that they are able to catch up with their peers and reach their full growth potential at a later age [67],[68]. We assess the possibility of such “catch-up” growth by running the analysis on sub-samples of older children. The results presented in Table 8 indeed provide some evidence in support of this. Above age three, excessive monsoon rainfall during infancy is no longer associated with child stunting. Similarly, above age one excessive monsoon rainfall in utero does not seem to affect the risk of stunting. However, these results might also be driven by selective survival, meaning that children who have been most affected by climate shocks in utero or during infancy are less likely to survive to later ages and would drop out from our sample. Further research is needed to determine the possibility of “catch-up” growth and selective survival, preferably with panel data.”

Overall Discussion Issues

20. Why did you see no effect for urban children? It is understandable why drought wouldnt matter -they are not producing their own food. But presumeably the effects of flooding on sanitation and infectious disease would matter in urban areas? Maybe use the DHS to add summary statistics about whether urban households have improved sanitation facilities, if the India DHS has that data.

Response: We have decided to remove from the analysis of urban-rural differences in climate-related undernutrition. While it is important to investigate whether such differences in exposure to climate shocks and level of vulnerability exist between urban and rural populations, we think this should be addressed in more details in a separate study, which could also look at access to health-care, reliance on agricultural income and other factors. While revising the paper, we have decided to focus on other aspects which explain the link between climate shocks and child nutrition. In particular, we have added diarrhea as a dependent variable. We have also constructed alternative measures of climate shocks, droughts and floods, and performed a number of robustness checks. Splitting the sample into urban and rural would make dichotomising climate shocks more difficult due to smaller sample sizes and reduced heterogeneity. 

21. Also, address why you saw an opposite effect (floods are good for nutrition) in mountains?

Response: We have added the following discussion (lines 510-516):

“…the mountainous region in northern India is relatively dry; Increased precipitation during the monsoon season implies improved access to clean water, as well as increased crop yield [73], both of which can benefit child health. Deficient water in the summer months has become a particularly pressing problem in the Indo- Gangetic plain; Decreasing snowcap is leading to a reduced water supply in the summer season and affecting poorly irrigated farmers [74]. This may explain the reduction in diarrheal infections and child stunting during periods of high precipitation in the mountainous regions.” 

22. I would add the caveat that floods are hard to detect with 9-month or 12-month SPEI. A high 12-month SPEI could indicate several mildly wet months, or one severe flood followed by eleven normal months, and while these two scenarios will have the same SPEI score, they could have very divergent effects on nutrition. I would like to see another analysis that addresses this, perhaps by using the count of months with 1-month SPEI greater than a threshhold. However, if this is not feasible it should be at least be discussed in the caveats section.

Response: In our analysis, we use 1-month SPEI values averaged over the monsoon months (June to September). It is true that this will smooth out extreme SPEI values if SPEI varied substantially during the same monsoon season. To address this issue, we have now generated separate dummy variables for droughts (SPEI ≤ -1.5) and floods (SPEI ≥ 1.5). See lines 291-295:

“In addition to using SPEI as a continuous variable in our model, we construct variables for droughts and floods based on the SPEI categories described in Table 1 above. Monthly SPEI values below or equal to -1.5 are categorised as a drought events and values above or equal to 1.5 are categorized as flood events. We then assess the effect of experiencing at least one drought or flood event in a given monsoon season on the risk of undernutrition and diarrheal diseases.”

The results are presented in Tables 3 to 5, and the corresponding text. 

We also tried to generate a count variable for the number of months when SPEI is above/below certain threshold, however, since we only consider exposure during the monsoon season, which is 4 months long, there are very few counts above 1 or 2. For this reason, we preferred to use the dummy variables mentioned above (at least one drought/flood event in a given monsoon season).

23. You mention the issue of migration - most DHS surveys have data on how long a household has been in at their current location, so you can exclude children that were not at that location when they were in utero or in their first year. You should definitely address this, or at least mention that the India DHS does not include this variable.

Response: We added a number of robustness checks, where we exclude people who had changed location since in-utero/birth, and those who are not permanent residents in the place where the interview took place. The tables can be found in S3 Appendix. The results were robust. See lines 368-375:

“A few robustness checks were performed in order to eliminate potential biases in our sample composition and model specification. First, we exclude from the sample children who are not permanent residents at the place where the interview took place. Second, we exclude children whose households have changed location between the time of exposure to the climate shock and the time of measurement. Third, an additional control variable for low birth weight is added to the model. We did not add this control variable in the main model since we expect that it would be correlated with climate conditions during the in-utero period. In all three cases, the results remain robust.”

24. Interpret not just the sign and significant but also the magnitude of SPEI changes. How does the increase in the odds of stunting associated with and SPEI of 2 compare with the increase in the odds of stunting associated with having a less educated mother or being born a twin? This is important to contextualize the potential impact of climate shocks.

Response: Thank you for this suggestion. We have tried to improve the analysis by including a comparison of the effect sizes. See for example line 332: 

“…These effects are not trivial; Increasing mother’s level of education from none to primary reduced the risk of stunting by a similar magnitude.”

And line 345:

“…The effect of floods on child stunting is comparable to living in a household without access to safe sanitation facility.”

COMMENTS BY REVIEWER 3

25. There are several papers that measure extreme weather shocks by amount of rainfall. The deviation of current year/season rainfall amount from the last 25-30 years average is used as proxy measure of droughts/floods. This paper uses another measure SPEI. It would be useful to have some discussion in the paper on comparative advantage of SPEI over rainfall-based definition of droughts/floods. Is SPEI better measure than rainfall shocks?

Response: Thank you for this comment, we added a discussion on the advantages of using SPEI to monitor droughts/floods instead of more simple rainfall-based measures. See lines 191-195:

“The SPEI index was developed in 2010 [42] and was originally intended for drought monitoring. It is an improvement on earlier drought indices as it allows using temperature along with rainfall data to measure the accumulation of water deficit/surplus. The index is increasingly used to monitor floods as well, especially when calculated at shorter timescales, such as 1 to 3 months [43],[44]. Floods are particularly difficult to forecast and show poor correlation with simple rainfall-based measures [45].”

26. Related to previous comments, policymakers identify districts based on amount of deficient rainfall and then roll out public polices to moderate the effects of negative rainfall. So from policy perspective, rainfall shocks seem more intuitive than SPEI. Therefore, I would like to authors to include discussion on the difference between SPEI/rainfall shocks and comment which measure is more useful for policy design.

Response: We have added a discussion on the advantages of using SPEI instead of simple rainfall-based measures to monitor droughts/floods (see our answer to the previous comment). We have also added that SPEI data can be easily accessed online and can be directly used to compare climate conditions across time and space, without additional standardisation being necessary. See lines 196-200: 

“The SPEI index has the potential to improve both drought and flood forecasting. It can serve policy makers, particularly in countries with high risk levels but low levels of preparedness to such disasters. Global gridded SPEI data can be accessed online (https://spei.csic.es/spei) or calculated with available software. Additionally, the index allows direct comparisons across time and space without the need to do additional standardisation.”

In addition, we constructed a measure of rainfall anomalies (deviation of monsoon season rainfall from the location-specific long-term mean) and used it as a robustness check. The effects on stunting remain significant, but the effect sizes are much smaller when we use the rainfall-based measure. This implies that SPEI captures climate shocks better than the rainfall-based measure. See lines 401-409:

“As an additional robustness check, we construct an alternative climate measure. We use the monthly rainfall data from the CRU TS 3.25 database, which was used to generate the SPEI index, and restrict it to the monsoon months (June to September). We then generate a variable of rainfall anomalies as a standard deviation change in monsoon season rainfall from the location-specific long-term mean (1970-2016). We run the main model with new measure of rainfall anomalies. The results confirm that excessive precipitation during in-utero and infancy increases the risk of stunting, however, the effect sizes are reduced almost by half. This implies that chronic undernutrition is more sensitive to variations in SPEI rather than the rainfall-based measure. Results tables for all robustness checks are available in S3 Appendix.”

27. In the current version, SPEI has been used as continuous variable. The authors motivated the paper by explaining the negative effects of droughts/floods on childhood outcomes. In the current analysis, 1 S.D. increase in SPEI does not mean whether it is a drought or flood. In that vein, I am suggesting if authors could conduct an additional analysis and create a binary indicator of floods/droughts using the SPEI index. For example, club SPEI ≤ -2 Extreme drought and -2 < SPEI ≤ -1.5 Severe drought as binary indicator of drought and similarly for flood.

Response: Thank you for this suggestion, we have now included measures of droughts and floods based on the SPEI classification. See lines 291-295:

“In addition to using SPEI as a continuous variable in our model, we construct variables for droughts and floods based on the SPEI categories described in Table 1 above. Monthly SPEI values below or equal to -1.5 are categorised as a drought events and values above or equal to 1.5 are categorized as flood events. We then assess the effect of experiencing at least one drought or flood event in a given monsoon season on the risk of undernutrition and diarrheal diseases.”

The results of this additional analysis are available in Tables 3, 4 and 5.

28. Do climate regimes map to SPEI ranges in Table 1?

Response: We added maps which show temperature, precipitation and SPEI differences across India. We also included figures which show monthly and yearly differences in temperature, precipitation and SPEI by climate zone in India. See Fig 3 and 4 and the corresponding text.

29. Why maternal age at birth is included as control and not the current age of mothers? Provide an explanation.

Response: We included current age of the mother as a control variable instead of the age at birth.

30. I would like the baseline specification to include some indicator of birth outcomes as control covariate, for example, I would suggest to include birthweight as covariate in the multivariate logistic model.

Response: We performed a robustness analysis where we included a variable for low birthweight in the baseline model. The results remained robust. See S12 Table in S3 Appendix and lines 396-399 in the manuscript:

“Third, an additional control variable for low birth weight is added to the model. We did not add this control variable in the main model since we expect that it would be correlated with climate conditions during the in-utero period. In all three cases, the results remain robust.”

31. Why age splines are included and not age in continuous months/years?

Response: Since child growth does not follow a linear pattern but is concentrated in the first months of life, age splines can capture better this non-linear growth trajectory. We use cubic age splines which are similar to polynomials of degree 3. We added the following explanation in the manuscript, line 283:

“The spline function fits polynomials of degree 3 between the defined knots in a way which ensures that levels and derivatives are equal on each side, and quadratic terms at each end…”

32. Access to sanitation in year 1 has been found to affect stunting, could authors include toilet access in the control set of variables?

Response: Thank you for this suggestion, we included type of sanitation facility to the set of control variables. We additionally included an interaction term between SPEI and the type of sanitation facility to see whether the lack of safe facility increases the risk of undernutrition due to excessive rainfall. The results are available in Table 7 and the corresponding text.

33. Line 262, results interpretation; “reports that 1 standard deviation (SD) increase in monsoon season SPEI in utero……”. Could we interpret this as “increase in drought…”? or is it possible to map this SPEI increase to drought? I am not familiar with SPEI literature so clarification on this would help the readers.

Response: We have tried to improve the interpretation of the results, providing more intuitive explanation. For example, see lines 325-332:

“Table 3, reports that an increase in the SPEI score of 1 during the in-utero period, which indicates wetter than usual monsoon season, reduces HAZ by 0.034 and increases the odds of stunting by 5% (both at 99% confidence, Table 3, col. 1 and 2). However, it is not clear whether this relationship is linear, it may be the case that both deficient and excessive precipitation during the monsoon season increases the risk of undernutrition. Looking at the effects of extreme climate events, we find evidence that the association is linear; Monsoon season droughts (SPEI≤-1.5) reduce the risk of stunting by 5%, whereas floods (SPEI≥1.5) increase it by 4% (both at 95% confidence, Table 3, col. 5).”

34. Line 350- Authors discuss several hypothesis about mechanism. One of the mechanisms could be incidence of diarrhea. The DHS survey contains information on diarrhea for the sample children. I would like authors to explore this a bit and run the baseline specification with diarrhea as an outcome.

Response: Thank you for this idea. We run the analysis with diarrhoea as an outcome variable and SPEI in the month of interview as the main variable of interest, since the DHS surveys only collect information on diarrhoea in the two weeks preceding the survey. In addition, we included an interaction between SPEI and the season of interview to detect potential seasonal patterns. Indeed, the results show that increase in SPEI raises the risk of contracting diarrhoea, particularly during the monsoon season. See Table 6 and the corresponding text in the manuscript, lines 410-418: 

“One of the mechanisms through which we expect climate shocks to affect child nutrition is the transmission of water-borne diseases, such as diarrheal infections. We investigate this mechanism by re-running the baseline model with diarrhoea as a dependent variable and climate in the month of interview as the main explanatory variable. The results presented in Table 6 show that indeed excessive precipitation in the month of interview is associated with an increased risk of contracting diarrhoea. We run the model again with an interaction term between the climate variable and the season of interview to test whether the latter moderates the effects of climate shocks on diarrhoea. Indeed, excessive rainfall during the monsoon months is associated with an increased risk of contracting diarrhoea, while no significant effects are found in the other seasons (Table 6, col.2).”

---

## [Decision Letter · Decision Letter 1]

20 Mar 2020

PONE-D-19-29467R1

Monsoon weather and early childhood health in India

PLOS ONE

Dear Ms. Dimitrova,

Thank you for submitting your manuscript to PLOS ONE. After careful consideration, we feel that it has merit but does not fully meet PLOS ONE’s publication criteria as it currently stands. Therefore, we invite you to submit a revised version of the manuscript that addresses the points raised during the review process.

Although the reviewers in general satisfied with the revised manuscript, there are some minors points needs to be addressed. Kindly address those points given by Reviewer 2. 

We would appreciate receiving your revised manuscript by May 04 2020 11:59PM. To enhance the reproducibility of your results, we recommend that if applicable you deposit your laboratory protocols in protocols.io, where a protocol can be assigned its own identifier (DOI) such that it can be cited independently in the future. For instructions see: http://journals.plos.org/plosone/s/submission-guidelines#loc-laboratory-protocols

We look forward to receiving your revised manuscript.

Kind regards,

Kannan Navaneetham

Academic Editor

PLOS ONE

Reviewers' comments:

Reviewer's Responses to Questions

**Comments to the Author**

1. If the authors have adequately addressed your comments raised in a previous round of review and you feel that this manuscript is now acceptable for publication, you may indicate that here to bypass the “Comments to the Author” section, enter your conflict of interest statement in the “Confidential to Editor” section, and submit your "Accept" recommendation.

Reviewer #2: All comments have been addressed

Reviewer #3: All comments have been addressed

2. Is the manuscript technically sound, and do the data support the conclusions?

Reviewer #2: Yes

Reviewer #3: Yes

3. Has the statistical analysis been performed appropriately and rigorously? 

Reviewer #2: Yes

Reviewer #3: Yes

4. Have the authors made all data underlying the findings in their manuscript fully available?

Reviewer #2: Yes

Reviewer #3: Yes

5. Is the manuscript presented in an intelligible fashion and written in standard English?

Reviewer #2: Yes

Reviewer #3: Yes

6. Review Comments to the Author

Reviewer #2: I think this manuscript is much improved and now makes a great contribution to the literature. It is stroner now that you have added subanalyses of including DD, an assessment of catch up growth, OLS regressions for HAZ & WHZ, and robustness checks for time at site.

Minor things:

209-210: For the benefit of researchers without a geophysical background, clarify why it is important that SPEI includes temperature: because temp influences evapotranspiration. Thus, in very warm, sunny conditions, excessive rainfall will quickly dry up, whereas in cooler, cloudy conditions, excessive water will stick around longer, potentially further affecting sanitary conditions. SPEI accounts for this in a way that rainfall alone does not.

446: This is confusing. Do you mean "a few studies"?

Finally, I would love to see some publicly available code for this analysis, perhaps as a github repository. This would be useful for future researchers interested in building on your results.

Reviewer #3: The authors have adequately addressed my comments and I have no further suggestions to make. I recommend acceptance.

7. PLOS authors have the option to publish the peer review history of their article (what does this mean?). If published, this will include your full peer review and any attached files.

Reviewer #2: Yes: Matthew Cooper

Reviewer #3: Yes: Santosh Kumar

---

## [Author Response · Author response to Decision Letter 1]

22 Mar 2020

COMMENTS BY REVIEWER 2

1. Minor things:

209-210: For the benefit of researchers without a geophysical background, clarify why it is important that SPEI includes temperature: because temp influences evapotranspiration. Thus, in very warm, sunny conditions, excessive rainfall will quickly dry up, whereas in cooler, cloudy conditions, excessive water will stick around longer, potentially further affecting sanitary conditions. SPEI accounts for this in a way that rainfall alone does not.

Response: Thank you for this suggestion. We have included the following text on lines 280-284: 

“It is important to account for the effect of temperature on evapotranspiration in addition to rainfall. For example, in sunny places with high temperatures, excessive rainfall will quickly dry up, whereas in cooler and cloudier places, excessive water will accumulate and remain for longer, potentially affecting sanitary conditions. SPEI accounts for this in a way that rainfall alone does not.”

2. 446: This is confusing. Do you mean "a few studies"?

Response: Yes, thank you for the correction.

3. Finally, I would love to see some publicly available code for this analysis, perhaps as a github repository. This would be useful for future researchers interested in building on your results.

Response: We are open to share the codes for producing this analysis on github and other platforms to facilitate future research on this topic.

---

## [Editor Report · Decision Letter 2]

25 Mar 2020

Monsoon weather and early childhood health in India

PONE-D-19-29467R2

Dear Dr. Dimitrova,

We are pleased to inform you that your manuscript has been judged scientifically suitable for publication and will be formally accepted for publication once it complies with all outstanding technical requirements.

With kind regards,

Kannan Navaneetham

Academic Editor

PLOS ONE
---

## [Editor Report · Acceptance letter]

27 Mar 2020

PONE-D-19-29467R2 

Monsoon weather and early childhood health in India 

Dear Dr. Dimitrova:

I am pleased to inform you that your manuscript has been deemed suitable for publication in PLOS ONE. Congratulations! Your manuscript is now with our production department. 

With kind regards,

on behalf of

Professor Kannan Navaneetham 

Academic Editor

PLOS ONE